# Upregulation of Transferrin Receptor 1 (TfR1) but Not Glucose Transporter 1 (GLUT1) or CD98hc at the Blood–Brain Barrier in Response to Valproic Acid

**DOI:** 10.3390/cells13141181

**Published:** 2024-07-11

**Authors:** Steinunn Sara Helgudóttir, Kasper Bendix Johnsen, Lisa Greve Routhe, Charlotte Laurfelt Munch Rasmussen, Maj Schneider Thomsen, Torben Moos

**Affiliations:** 1Neurobiology Research and Drug Delivery (NRD), Department of Health Science and Technology, Aalborg University, 9260 Gistrup, Denmark; ssh@2npharma.com (S.S.H.); ljrouthe@gmail.com (L.G.R.); clmrasmussen@health.sdu.dk (C.L.M.R.); 2Section for Biotherapeutic Engineering and Drug Targeting, Department of Health Technology, Technical University of Denmark, 2800 Lyngby, Denmark; kasperbendixjohnsen@gmail.com

**Keywords:** blood–brain barrier, capillary depletion, CD98hc, cell culture, glucose transporter 1, ICP-MS, nanoparticle, transferrin receptor, valproic acid

## Abstract

Background: Transferrin receptor 1 (TfR1), glucose transporter 1 (GLUT1), and CD98hc are candidates for targeted therapy at the blood–brain barrier (BBB). Our objective was to challenge the expression of TfR1, GLUT1, and CD98hc in brain capillaries using the histone deacetylase inhibitor (HDACi) valproic acid (VPA). Methods: Primary mouse brain capillary endothelial cells (BCECs) and brain capillaries isolated from mice injected intraperitoneally with VPA were examined using RT-qPCR and ELISA. Targeting to the BBB was performed by injecting monoclonal anti-TfR1 (Ri7217)-conjugated gold nanoparticles measured using ICP-MS. Results: In BCECs co-cultured with glial cells, *Tfrc* mRNA expression was significantly higher after 6 h VPA, returning to baseline after 24 h. In vivo *Glut1* mRNA expression was significantly higher in males, but not females, receiving VPA, whereas *Cd98hc* mRNA expression was unaffected by VPA. TfR1 increased significantly in vivo after VPA, whereas GLUT1 and CD98hc were unchanged. The uptake of anti-TfR1-conjugated nanoparticles was unaltered by VPA despite upregulated TfR expression. Conclusions: VPA upregulates TfR1 in brain endothelium in vivo and in vitro. VPA does not increase GLUT1 and CD98hc proteins. The increase in TfR1 does not result in higher anti-TfR1 antibody targetability, suggesting targeting sufficiently occurs with available transferrin receptors without further contribution from accessory VPA-induced TfR1.

## 1. Introduction

While denoting the primary obstacle for delivering therapeutics to the brain [1,2], the blood–brain barrier (BBB) is effective in maintaining a stable extracellular environment and prevents unwanted entry of toxins, pathogens, and other harmful molecules [1,3,4]. The BBB is formed by a monolayer of highly specialized brain capillary endothelial cells (BCECs) connected by their intermingling tight junctions, which limit non-specific, paracellular transport [4,5,6,7,8]. Crucial nutrients entering the brain undergo active transport via receptors and transporters expressed on the luminal surfaces of the BCECs, some of which also have an interest for delivering targeted therapeutics to the brain [5,9].

Limited understanding of expressional patterns and functional regulation of these receptors and transporters remains a challenge for the development of targeted drugs [10,11]. Drug delivery strategies often target endogenously expressed nutrient transporters delivering, e.g., iron (transferrin receptor 1 (TfR1)), glucose (glucose transporter 1 (GLUT1)), or large amino acids (Cluster of Differentiation 98 Heavy Chain (CD98hc). Their availability is under the influence of nutritional demand, age, inflammation, and other pathological processes [12]. Most drug delivery efforts have focused on TfR1; however, it has proven rather complicated to design the optimal construct for maximum brain uptake and retention [13]. GLUT1 is an attractive target due to its abundance in the BBB and seems to mediate transport through a carrier-mediated pathway that does not rely on a specific receptor but rather a solute carrier protein [14]. CD98hc has also proven an interesting target consisting of the heavy chain of large amino acid transporter. It seems to be expressed both on the luminal and abluminal plasma membrane of BCECs [13,15]. 

The expression of nutrient transporters is probably also challenged by pharmacological treatment, and little is known about, e.g., the possible responses of BCECs to treatment with valproic acid (VPA), which is used for the treatment of bipolar disorders and epilepsy. VPA is thought to induce epigenetic changes by inhibiting histone deacetylase, which further enables acetylation of histones and consequently gene transcription [16]. 

As we have previously shown that the expression of ferroportin, another iron-transporting molecule (aka SLC40A1 or IREG1), is genetically upregulated in isolated rat BCECs in response to VPA treatment [17], we hypothesize that VPA can induce changes in BCECs that modify the expression of important drug delivery targets, i.e., TfR1, GLUT1, and CD98hc for drug delivery to the brain [18]. We report that VPA treatment in mice upregulates TfR1 at the BBB in vivo when examining isolated brain capillaries and in vitro when studying isolated primary endothelial cells. In contrast, VPA treatment does not influence the expression of GLUT1 and CD98hc.

## 2. Material and Method

The following reagents were purchased from Sigma/Millicell Merck KGaA (Darmstadt, Germany, DE): Insulin transferrin sodium selenite (Cat. no. 11074547001), puromycin (Cat. no. P8833), collagen type IV (Cat. no. C5533), fibronectin (Cat. no. F1141), poly-L-lysine (Cat. no. P6282), hydrocortisone (HC) (Cat. no. H4001), dimethyl sulfoxide (DMSO) (Cat. no. D2650), 8-(4-chlorophenylthio)adenosine 3′,5′-cyclic monophosphate sodium salt (CTP-cAMP) (Cat. no. C3912), 4-(3-butoxy-4-methoxybenzyl)imidazolidin-2-one (RO) (Cat. no. B8279), paraformaldehyde (PFA) (Cat. no. 441244), triton x−100 (Cat. no. X100), 4′,6-diamidino−2-phenylindole dihydrochloride (DAPI), β-mercaptoethanol (Cat. no. M6250), cOmplete Mini, EDTA-free (Cat. no. 11836170001), valproic acid sodium salt (Cat. no. P4543), dextran (MW 60,000 Da), and percoll (Cat. no. P4937).

The following reagents were purchased from Thermo Scientific (Nærum, Denmark): Fetal calf serum (FCS) (Cat. no. 10270), Dulbecco’s Modified Eagle Medium consisting of nutrient mixture F-12 (DMEM/F-12) (Cat. no. 31331), DMEM (low glucose) (Cat. no. 21885), DMEM (high glucose) (Cat. no. 31966), trypsin (Cat. no. 15090-46), phosphate-buffered saline (PBS) (Cat. no. SH3025802), rabbit anti-zonula occludens 1 (ZO-1) (Cat. no. 61-7300), Alexa Fluor 488-conjugated goat anti-rabbit IgG (Cat. no. A11034), Alexa Fluor 594-conjugated goat anti-rat IgG (Cat. no. A11007), RevertAid H Minus First Strand cDNA Synthesis Kit (Cat. no. K1652), DNase I enzyme (Cat. no. EN0521), TaqMan Multiplex MasterMix (Cat. no. 4484262), N-PER neuronal protein extraction reagent (Cat. no. 87792), BCA protein assay kit (Cat. no. 23225), TRIzol Plus RNA Purification Kit (Cat. no. 12183555), Taqman Fast Advanced MasterMix (Cat. no. 4444963), N-PER neuronal protein extraction reagent (Cat. no 87792), BCA protein essay kit (Cat. no. 23225), and Taqman Probes for CD98hc (Cat. no. 4331182), Glut1 (Cat. no. 4331182), and Hprt1 (Cat. no. 4448490).

AllPrep DNA/RNA Mini Kit (Cat. no. 80204) was purchased from Qiagen (Hilden, Germany). Greiner Bio-one (Kremsmünster, Austria) Thincert cell culture inserts for 12-well plates with a transparent polyethylene terephthalate (PET) membrane and a pore diameter of 1 µm (Cat. no. 665610) were purchased from In Vitro (Fredensborg, Denmark). Basic fibroblast growth factor (bFGF) (Cat. no. 100-18B) was purchased from PeproTech Nordic (Stockholm, Sweden). Gentamicin sulfate (Cat. no. 17-518Z) was purchased from Lonza Copenhagen (Vallensbaek Strand, Denmark). Plasma-derived bovine serum (PDS) (Cat. no. 60-00-810) was purchased from First Link (Wolverhampton, UK). Bovine serum albumin (BSA) (Cat. no. EQBAH62) was purchased from Europa Bioproducts (Cambridge, UK). Fluorescence mounting medium (Cat. no. S3023) was purchased from DAKO (Glostrup, Denmark). Mouse TfR1 ELISA kit (Cat. no. EKM2800) was purchased from Nordic BioSite (Copenhagen, Denmark). Isoflurane was purchased from Baxter A/S (Søborg, Denmark). Collagenase Type II was purchased from Life Technologies. Collagenase/dispase was purchased from Roche.

### 2.1. Isolation of BCECs

Mice were purchased from Janvier labs (Le Genest-Saint-Isle, France). Primary brain capillary endothelial cells (BCECs) and pericytes were isolated from eight-week-old C57BL/6 mice as previously described [19,20]. The mice were anesthetized using isoflurane and their heads submerged in 70% ethanol before decapitation. The meninges and the brain’s cortical visible white matter were removed, and the remaining brain tissue was immersed in cold DMEM-F12 medium. They were then digested in DMEM-F12 medium supplemented with collagenase type II and DNase I enzyme at 37 °C for 75 min in an Incubated mini Shaker (VWR, Søborg, Denmark). Cells were then separated by mixing 20% bovine serum albumin (BSA) into the solution, followed by a centrifugation step. The pellet containing the microvessels was further digested with collagenase/dispase and DNase I enzyme for 50 min at 37 °C and separated on a 33% percoll gradient. Isolated BCECs were maintained in T75 flasks precoated with collagen IV and fibronectin in endothelial medium, which consisted of DMEM-F12 supplemented with 10% PDS, 0.1 mg/mL ITS, 10 μg/mL gentamicin sulfate, and freshly added 1 ng/mL bFGF. In order to remove pericyte contamination, the BCECs were cultured in media supplemented with 4 µg/mL puromycin for four days [21,22].

### 2.2. Isolation of Mixed Glia Cells

Mixed glial cells were isolated as previously described [20]. The cells were resuspended in DMEM-F12 GlutaMAX medium supplemented with 10% FCS and 10 μg/mL gentamicin sulfate and seeded in flasks pre-coated with poly-L-lysine. The mixed glial cells were cultured for three weeks in an incubator at 5% CO_2_ and 37 °C. The cells were then frozen for later use in DMEM supplemented with 30% FCS and 7.5% DMSO. The cells were then seeded in poly-L-lysine coated 12-well plates and cultured 10 to 14 days until confluence.

### 2.3. Construction of an In Vitro Co-Culture BBB Model

The BCECs were cultured in a CO_2_ incubator with 5% CO_2_ and 95% O_2_ at 37 °C. They were then passaged to 12-well hanging culture inserts precoated twice with 1 mg/mL collagen IV and 1 mg/mL fibronectin diluted in Milli-Q water at a density of 100,000 cells/cm^2^. Once the hanging insert culture contained a fully confluent monolayer of BCECs, they were placed in a 12-well plate containing a population of mixed glia cells, mainly astrocytes. The medium was further supplemented with tight-junction-inducing factors, where the medium in the top compartment consisted of 550 nM HC, 250 μM CTP-cAMP, and 17.5 μM Ro 20-1724 and the medium in the bottom compartment consisted of a 1:1 mixture of endothelia medium and astrocyte condition medium supplemented with 550 nM HC. The integrity of the BCEC monolayers was evaluated daily by measuring the transendothelial electrical resistance (TEER) using a Millicell ERS-2 epithelial volt-ohm meter and STX01 Chopstick Electrodes (Merck Millipore, Hellerup, Denmark). The measured TEER was subtracted from the value of an empty hanging insert containing culture medium and the value was multiplied by the area of the culture insert. The calculated TEER values were presented as mean Ω × cm^2^ ± SD. Once the barrier was tight, half of the barriers were treated with 2 mM VPA added directly to the top chamber for 6 h or 24 h.

### 2.4. In Vivo Studies

The mice were processed as shown in Figure 1. Animal experiments were approved by the Animal Experiments Inspectorate under the Danish Ministry of Environment and Food (license number: 2018-15-0201-01550) and performed under the European Legislation of Animal Experimentation 2010/63/EU. A total of 48 C57BL/6JRj mice aged 7 weeks, 16–24 g, were used for the study. To account for possible gender differences in the response to VPA [23,24], both male (*n* = 24) and female (*n* = 24) mice were included. The mice were purchased from Janvier Labs (Le Genest-Saint-Isle, France), fed with a commercial diet (Altromin 1324, Brogaarden, Denmark), and had access to food and water ad libitum. They were housed in cages of five with all experimental groups represented in each cage and acclimatized to the local environment (temperature 20–22 °C, humidity 40–60%, and light-dark cycle of 12 h) at the animal facility of Aalborg University for 14 days before experiments.

### 2.5. Labeling of Anti-TfR1 Antibodies with 1.4 nm Gold Nanoparticles

To estimate the surface availability and transport capability of TfRs at the BBB, we used anti-TfR1 antibodies (clone Ri7217, produced in-house using the hybridoma technique) [25] labelled with 1.4 nm gold nanoparticles (Nanoprobes Inc., Yaphank, NY, USA) for high sensitivity detection with inductively-coupled plasma-mass spectrometry. The stock buffer of the anti-TfR1 antibodies (PBS) was exchanged to 0.1 M sodium borate buffer with 2 mM EDTA (pH 8). The buffer-exchanged antibodies were then subjected to thiolation using Traut’s reagent (Thermo Scientific, Hvidovre, Denmark) at a reagent-to-antibody ratio of 10. The solution was allowed to incubate for one hour at room temperature under constant shaking at 500 rpm. The thiolated antibodies were then transferred to an Amicon Ultra spin filter (MW cut-off 30 kDa) (Sigma-Aldrich, Søborg, Denmark), topped with 5 mL sodium borate buffer, and centrifuged at 4000× *g* for 20 min at room temperature. The filter-through was discarded, and 5 mL PBS (pH 7.4) was added to the thiolated antibodies. After volume reduction to 50 µL by centrifugation, the thiolated antibodies were transferred to a Protein Lo-Bind Eppendorf tube and stored at 4 °C for no more than 30 min. The conjugation of thiolated antibodies to the 1.4 nm gold nanoparticles was performed using Monomaleimide Nanogold Labeling Reagent (Nanoprobes Inc., Yaphank, NY, USA) according to the manufacturer’s instructions. In short, the lyophilized monomaleide nanoparticles were mixed in 1 mL deionized water and added to 2 mg thiolated antibodies. The air phase was replaced with N_2_, and the solution was incubated at 4 °C overnight. After incubation, the labeled antibodies were separated from unbound gold nanoparticles using gel filtration chromatography on a Superose 6 column (Sigma-Aldrich, Søborg, Denmark) and stored at 4 °C until use.

### 2.6. VPA Administration In Vivo

Mice were randomized into three groups based on body weight. VPA was diluted in sterile PBS and injected intraperitoneally (i.p.) in doses of 100 mg/kg or 400 mg/kg, previously deemed suitable for mice [26]. The control group received injections of sterile PBS. Mice were euthanized 24 h post-injection. Animals were weighed regularly and monitored for their well-being up until two hours post-injection. They were allowed to recover in a heated cage post-injection. VPA injected in a dose of 100 mg/kg induced a phenotype where the mice became ataxic shortly after administration. In contrast, mice dosed with 400 mg/kg were heavily sedated, which lasted for up to 1 h; after recovery, they presented an ataxic phenotype but were capable of moving freely in their cage. The adverse effects disappeared within 24 h, leading to the full recovery of all mice before termination.

### 2.7. Uptake at the BBB of Nanogold-Conjugated Anti-TfR1 Antibodies

Mice (*n* = 12) were intravenously injected with nanogold-conjugated anti-TfR1 antibodies (clone Ri7217) 24 h after 100 mg/kg VPA treatment (*n* = 6) or injection of PBS (*n* = 6) (see below). The animals were monitored for 60 min after the nanogold-conjugated anti-TfR1 was injected and then euthanized by transcardial perfusion with 20 mL 0.01 M KPBS (pH 7.4) under isoflurane anesthesia; in these mice, the brains and peripheral organs of interest were dissected, snap frozen in liquid nitrogen, and kept at −80 °C until further analysis.

### 2.8. Immunocytochemistry

Immunocytochemistry was carried out on BCECs to investigate the presence of TfR1 following VPA treatment. Cells were also stained for the tight-junction-associated protein ZO-1 to investigate if the cellular integrity was affected by the treatment. The medium was harvested and the BCECs washed in 0.1 M PBS, fixed for 10 min in 4% PFA, and then washed twice in PBS. To block the unspecific binding of antibodies, the cells were incubated for 30 min in 3% BSA and 0.3% Triton X-100 diluted in 0.1 M PBS. BCECs were incubated for 60 min with anti-TfR1 (clone Ri7217) and ZO-1 diluted 1:200 in a blocking buffer. The cells were subsequently washed twice in a washing buffer to remove any unbound primary antibodies. The cells were then incubated for 60 min with the secondary antibodies Alexa Flour 594-conjugated goat anti-rat and Alexa Flour 488-conjugated goat anti-rabbit diluted 1:200 in blocking buffer followed by washing. The nuclei were stained with DAPI diluted 1:1000 in PBS for 5 min. Cells were mounted onto slides using DAKO fluorescent mounting medium and examined using an AxioObserver Z1 fluorescence microscope equipped with an ApoTome.2 and Axiocam MR camera under a Plan-Apochromat 40 ×/1.3 NA oil DIC objective. Images were corrected for brightness and contrast using Fiji ImageJ2.

### 2.9. Capillary Depletion

Brain capillary isolation was performed to separate the brain capillaries from the remaining brain tissue. Brains from mice dosed with 100 mg/kg VPA, 400 mg/kg VPA, or PBS were dissected, and the right cerebral hemisphere was sampled and transferred to a Dounce homogenizer. Ice-cold capillary depletion buffer [27] was added, and the brains were homogenized by six strokes, followed by mixing with ice-cold 30% dextran and further homogenization with six strokes. The homogenate was transferred into a 15 mL Falcon tube and centrifuged at 5400× *g* for 40 min at 4 °C. The capillary and supernatant fractions were carefully separated and stored at –80 °C until further analyses.

### 2.10. Quantification of Tissue Gold Content

The gold content in the tissue samples was analyzed using inductively coupled plasma mass spectrometry. The samples were digested in aqua regia overnight at 65 °C, followed by dilution in deionized water containing 0.5 ppb iridium (Fluka, Sigma-Aldrich, Brøndby, Denmark). Immediately before analysis, the samples were diluted in 2% HCl containing 0.5 ppb iridium, after which they were analyzed on an iCAP Q ICP-MS system (Thermo Scientific, Hvidovre, Denmark) fitted with an ASX-520 AutoSampler and a Neclar ThermoFlex 2500 chiller. Performance on the instrument was ensured by calibration using TUNE B iCAP Q element mixture (Thermo Scientific, Hvidovre, Denmark). A standard curve was generated by serial dilution of an analytical grade gold standard solution to obtain the gold concentration in each sample (Fluka, Sigma-Aldrich, Brøndby, Denmark). Measurements of the iridium concentration were used as an internal standard to ensure similar analysis for all samples.

### 2.11. Probe-Based RT-qPCR

The purification of RNA from BCECs was carried out using the AllPrep DNA/RNA Mini kit following the manufacturer’s protocol. RNA concentrations were assessed using a DS-11 FX Spectrophotometer/Fluorometer (DeNovix, Wilmington, NC, USA). A total of 150 ng RNA was treated with DNAse I enzyme to remove any potential genomic contamination, mixed with nuclease free water and 10× reaction buffer and incubated for 30 min in a T100 Thermal cycler (Bio-rad) at 37 °C. All samples were then treated with Ethylenediaminetetraacetic acid (EDTA) for 10 min at 65 °C to stop the reaction. Complementary DNA was generated using the RevertAid H Minus First Strand cDNA Synthesis Kit, which consisted of random hexamer primer, oligo primer, 10 mM dNTP, 5× reaction buffer, and nuclease free water. This was mixed with Maxima H Enzyme mix and 100 ng of DNase-treated RNA and the samples run in the thermal cycler using the following thermal profile: 10 min at 25 °C, 15 min at 50 °C, and 5 min at 85 °C. The RT-qPCR was carried out by preparing a master mix containing the TaqMan Multiplex MasterMix and the Taqman primers/probes for *Glut1*, *CD98hc*, *Tfrc*, and *Hprt1* and then adding 1.7 ng of sample into each well. The samples were run on a QuantStudio 6 Flex Real-Time PCR System (Thermo Fisher Scientific) with the thermal profile: 50 °C for 2 min and 95 °C for 20 s, followed by 40 cycles of 95 °C for 1 s and 60 °C for 20 s. After the run, the samples were cooled down to 20 °C. The relative mRNA expression was calculated using the ∆∆CT method, and the expression was normalized to *Hprt1* expression, where PBS-injected mice served as the reference sample.

### 2.12. ELISA

Enzyme-linked immunosorbent assay (ELISA) was used to measure the amount of TfR1, GLUT1, and CD98hc in the isolated brain capillaries after PBS or VPA treatment to evaluate these proteins at the level of the BBB [28]. To account for varying protein concentrations in each sample, the total protein concentration was measured using a BCA protein assay kit as previously described [17]. Each sample was lysed using 90 µL N-PER^TM^ Neuronal protein extraction buffer supplemented with cOmplete Mini protease inhibitor. The samples were lysed on ice for 10 min, spun down at 14,000× *g* for 10 min at 4 °C, and the supernatants collected. Samples were diluted 1:10 in sample buffer and analyzed using sandwich ELISA for mouse TfR1, GLUT1, and CD98hc according to the manufacturer’s protocol. The absorbance was read at 450 nm using an Enspire Plate Reader from Perkin Elmer, and the concentration was calculated using a standard curve and normalized to the total protein concentration of each sample.

### 2.13. Statistical Analysis

The relative gene expression of *Tfrc*, *Glut1*, and *Cd98hc* mRNA from mice and isolated BCECs was calculated according to the ∆∆CT method. All data were checked for a normal distribution before performing statistical analysis. The results were analyzed using one-way ANOVA with Dunnett’s multiple comparisons test in GraphPad Prism 9. When investigating the difference in mRNA expression between male and female mice, a two-way ANOVA with the Holm–Sidak multiple comparisons test was used. ELISA data on TfR1, GLUT1, and CD98hc in brain capillaries were analyzed by one-way ANOVA with Dunnett’s multiple comparisons test. The luminal binding of Nanogold-conjugated anti-TfR1 antibodies was analyzed by unpaired *t*-test. The significance levels were * *p* = 0.01–0.05, ** *p* = 0.001–0.01, *** *p* = 0.0001–0.001, and **** *p* < 0.0001.

## 3. Result

### 3.1. VPA Administration Upregulates Tfrc in BCECs In Vitro

To evaluate the effects of VPA on the expression of important drug delivery targets, BCECs co-cultured with mixed glial cells were treated with 2mM VPA for 6 h or 24 h (Figure 2A). The integrity of the barriers was evaluated via measurement of TEER followed by immunolabeling of the tight-junction-associated protein ZO-1. The TEER was unaffected by VPA treatment both after 6 h and 24 h (Figure 2B). The expression and subcellular distribution of ZO-1 were similar between the control barriers and the barriers treated with VPA for 24 h (Figure 3). Having established that the VPA treatment did not affect the integrity of the in vitro barriers, we then analyzed the effect of VPA treatment on the expression profiles of *Tfrc*, *Glut1*, and *Cd98hc*. The mRNA expression of *Tfrc* was significantly higher after VPA treatment for 6 h, and the expression level returned to baseline after 24 h (Figure 2C). Conversely, the mRNA expression of *Glut1* and *Cd98hc* was unaffected by VPA treatment (Figure 2D,E). VPA did not only increase the gene expression of *Tfrc*. By visual inspection, the TfR1 protein was particularly high in the perinuclear region of the BCECs compared to the control BCECs, suggestive of an increased presence of TfR1 in the granular endoplasmic reticulum (Figure 3).

### 3.2. VPA Induces TfR1 Expression In Vivo

We next wanted to investigate if VPA had the same effects on the gene and protein expression of TfR1, GLUT1, and CD98hc in living animals as in the in vitro cultured BCECs. Analyzing the isolated brain capillaries from PBS, 100 mg/kg VPA, and 400 mg/kg VPA treated animals, the expression of *Tfrc* was unaltered between treatment groups 24 h post-injection (Figure 4A), with no difference in expression between male and female mice (Figure 4B). In contrast, the TfR1 protein in capillaries increased significantly after treatment with both 100 mg/kg and 400 mg/kg VPA (Figure 4C). The TfR1 level ranged from 1.5 to 3.7 ng/µg protein and was not impacted by the gender of the animals (Figure 4D).

### 3.3. Uptake of Gold-Labeled Anti-TfR1 Antibodies 

Next, we wanted to study whether the increased TfR1 protein in brain capillaries induced by VPA could be quantitatively reflected by increased uptake of gold-labeled anti-TfR1 antibodies. We administered the antibodies intravenously at a dose of 1 mg/kg to VPA-treated (100 mg/kg for 24 h) and control male mice and allowed them to circulate for 60 min. Using ICP-MS-based quantification of the gold content, the accumulation in brain capillaries was non-significant, suggesting that VPA treatment did not improve the uptake of the anti-TfR1 antibodies despite increased TfR1 (Figure 4E). Similarly, the uptake of the gold-labeled anti-TfR1 antibodies in liver, lung, and spleen, organs where TfR1 is mainly expressed in hepatocytes, and in macrophages, was unaltered (Figure 4F–H).

### 3.4. VPA Increases Glut1 mRNA in Male but Not in Female Mice

The relative gene expression of *Glut1* in isolated capillaries was unaffected by treatment with 100 mg/kg and 400 mg/kg VPA after 24 h (Figure 5A). However, gender-related differences were observed, as male mice treated with 400 mg/kg VPA for 24 h significantly upregulated their expression of *Glut1* compared to the PBS-treated male mice (Figure 5B). The increased expression of *Glut1* among the 400 mg/kg VPA treated male mice was not reflected in the GLUT1 protein (Figure 4D), as the GLUT1 protein content was unchanged in all treatment groups (Figure 5C,D). The content of the GLUT1 protein was in the range 0.4–1.8 ng/µg protein.

### 3.5. CD98hc Expression Is Not Affected by VPA Treatment

The relative gene expression of *CD98hc* was unaltered between the PBS- and VPA-treated mice, and no difference was observed between male and female mice (Figure 5E–H). The CD98hc protein was present in capillaries, ranging from 0.4 to 3.7 ng/µg protein.

## 4. Discussion

Targeting nutrient receptors and transporters that are highly expressed at the BBB is of interest for the delivery of large therapeutics to the brain [29]. The intriguing possibility of pharmacologically increasing the expression of TfR1, GLUT1, or CD98hc on BCECs in a manner that could increase the availability of targetable molecules inspired this study.

VPA is known to interact at the level of the BBB, e.g., VPA prevents BBB disruption after subarachnoid hemorrhage [30] and intracerebral hemorrhage [31] and lessens BBB disruption after transient focal cerebral ischemia [32]. VPA inhibits the histone deacetylation process, thereby ensuring the accessibility of transcription factors to the chromatin, which increases the gene transcription of multiple genes and is therefore not specific to the aforementioned targets [16,32,33]. VPA was initially approved as an anti-epileptic drug, but its ability to inhibit HDAC could explain the increase in TfR1 observed in the current study [34].

TfR1 remains the most-studied receptor for targeting the BBB. The reason for its popularity is that TfR1 is only expressed on BCECs and not on other endothelial cells in the body [35], which leads to a preferential cerebral accumulation of TfR1-targeted molecules [35,36]. TfR1 promotes the uptake of transferrin-bound iron, which is an essential co-factor in multiple physiological functions such as cell division, DNA synthesis, and oxygen transport [37,38,39,40,41]. We found that VPA treatment increases the expression of *Tfrc* significantly after 6 h in vitro, reaching an expression of three times greater than the controls, and that the expression was back to baseline after 24 h of treatment. The time-dependent upregulation of *Tfrc* may be explained by a negative feedback loop. To examine whether this effect was species specific, we have conducted similar experiments in the rat where *Tfrc* was increased 2.5-fold following VPA treatment [42].

The relative gene expression of *Tfrc* was also investigated in purified brain capillaries from mice treated with VPA for 24 h. However, there was no indication of altered expression of *Tfrc* 24 h post-injection, but TfR protein levels were upregulated with VPA treatment. It remains to be answered whether the expression of *Tfrc* is regulated by post-transcriptional mechanisms rather than regulation of mRNA synthesis [43], meaning that in conditions with, e.g., deprivation of nutritional iron supply, cells upregulate the TfR1 protein by increasing translation of its mRNA.

In BCECs, the TfR1content is particularly high during the development of the brain and ceases with increasing postnatal age [44,45]. In the rat, this was also reflected in a developmentally higher uptake of iron-containing transferrin and a monoclonal antibody (OX26) targeting TfR1 following intravenous injection [46]. The current study suggests that VPA also leads to increased post-translational regulation of TfR1 in BCECs. Prior studies in isolated porcine BCECs suggest that BCECs harbor a significant spare pool of transferrin receptors that can be mobilized when needed, as occurring in conditions with cerebral iron deficiency [47]. The present study examining adult mice did not record an increased uptake of 1.4 nm gold-conjugated anti-transferrin receptor antibodies despite a higher content of TfR1 following VPA treatment. Injection of high-affinity anti-TfR1 leads to a rapid satiation of the TfR1, peaking by 1 h after injection with a level of −0.6% of the injected dose [48] and without significantly higher uptake after longer exposure [49]. The absence of a higher uptake following VPA treatment suggests that the quantity of functionally available TfR1 molecules contributing to the cellular uptake of the targeted anti-TfR1 antibody remained unaltered in BCECs following VPA treatment. This also suggests that the observed increase in TfR1 protein in BCECs following VPA treatment is attributed to an enlargement of the intracellular TfR1 pool. Possibly, the BCECs’ uptake of iron-transferrin and targeted antibodies in adults operates primarily via mobilization of this intracellular pool of spare TfR1, which could acquire a contribution from the pharmacologically enhanced TfR1 protein.

GLUT1 in the brain is selectively confined to BCECs, among the most abundant proteins in BCECs, and regulated in accordance with the brain’s metabolic needs [50,51,52]. A reduction in GLUT1 has been observed in Alzheimer’s disease, making it less suitable for drug delivery [53,54,55,56,57]. The impact of VPA on glucose metabolism has been reported before. This is underscored by the clinical guideline that VPA is contraindicated for epilepsy patients with GLUT1 deficiency syndrome [58].

Both in vivo and in vitro experiments have demonstrated the effects of VPA on glucose metabolism. VPA treatment (2 × 300 mg/kg IP daily) for seven consecutive days selectively reduced the brain Glut1 mRNA expression in male rats [59]. Similar results were found in primary astrocytes, where a 60% reduction of Glut1 mRNA was reported [60].

The current study does not see an overall alteration in *Glut1* when analyzing male and female mice together. However, gender-specific analysis reveals a significantly higher expression of *Glut1* in the brain capillary endothelial cells (BCECs) of male mice treated with VPA, an effect not seen in females. This increase in mRNA expression in BCECs contrasts with previous reports in whole brain tissue and primary cultured brain cells. It is speculated that this may be a compensatory mechanism, where the BBB enhances glucose import in response to compromised glucose metabolism within the brain. The reason for the differential impact of VPA on the acetylation status of the GLUT1 promoter in male mice remains unknown. These findings emphasize the importance of including both male and female subjects in research and distinguishing between different brain structures or cell types when discussing glucose metabolism.

Manipulation of histone acetylation affects around 2% of genes [16], and CD98hc seems to fall into the 98% of genes unaffected by acetylation status. CD98hc has been verified as a strong candidate for pharmacological targeting to the BBB [61], where various delivery platforms utilizing its high and selective expression are being investigated. CD98hc is an intracellular amino acid transporter and integrin signaling enhancer [18] and among the most highly expressed transcripts in human and mouse brain capillaries [29,36,62], which is in agreement with the high CD98hc protein level detected in the ELISA experiments in the present study.

The nutrient transporters on the BBB are dynamically regulated, and it is important to understand the up and downregulation of these when designing transport modalities for larger molecular constructs that deliver drugs to the brain. However, it is even more important to understand what impacts the activity of the transporter.

The regulation of nutrient transporters at the BBB is dynamic, necessitating a thorough understanding of their upregulation and downregulation in the context of developing transport modalities for large molecular constructs aimed at delivering drugs to the brain. Furthermore, elucidating the factors that influence the activity of these transporters is essential, as their activity ultimately determines the amount of drug delivery.

## 5. Conclusions

The current study showed that it is possible to upregulate the TfR1 protein in BCECs utilizing the histone deacetylase inhibitor VPA. Upregulation of *Tfrc mRNA* was seen in BCECs in vitro and *Glut1* mRNA increased in the BBB following VPA treatment in male mice. Upregulation of the GLUT1 or CD98hc proteins was not seen. The increase in the TfR1 protein at the BBB was not accompanied by higher targetability of a specific anti-TfR1-targeted monoclonal antibody, which suggests that targeted delivery to TfR1 engages with already available transferrin receptor without the further aid of VPA-induced TfR1.

## Figures and Tables

**Figure 1 cells-13-01181-f001:**
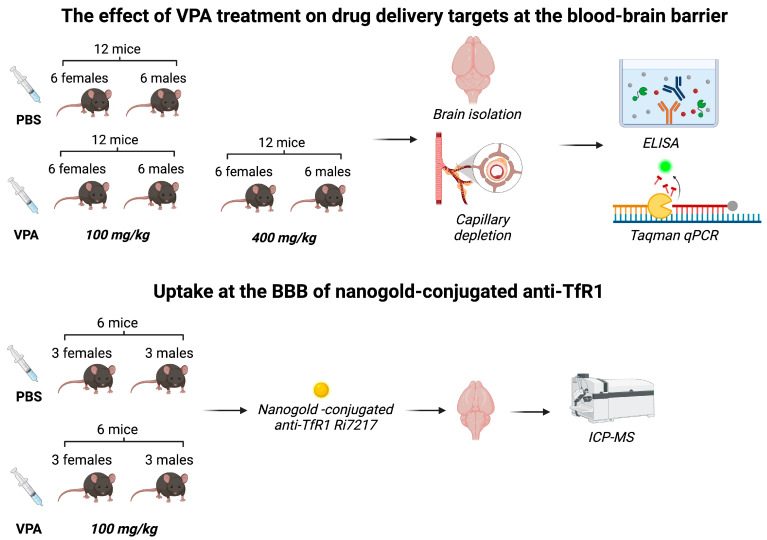
Flowchart showing experiments involving living mice.

**Figure 2 cells-13-01181-f002:**
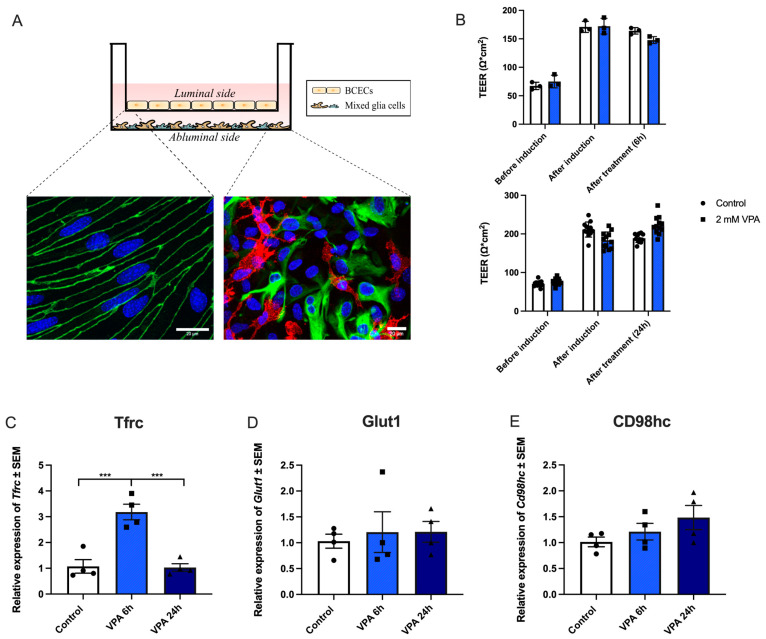
In vitro experiments addressing the effects of valproic acid (VPA) on gene expression of TFRfrc, Glut1, and Cd98hc. (**A**) Transwell system illustrating a non-contact co-culture system where brain capillary endothelial cells (BCECs) are cultured on hanging inserts, making up the luminal side of the barrier. The mixed glial cells are cultured in the bottom of the well, making up the abluminal side of the barrier. The immunocytochemistry illustrations display BCECs (on the left) stained with ZO-1 and Dapi and the mixed glia culture (on the right) stained for GFAP (green) to identify astrocytes and Cd11b (red) to identify microglia, as well as Dapi. The scale bar is 20 µm. (**B**) The transendothelial electrical resistance (TEER) was investigated before and after tight junction induction, as well as treatment with 2 mM VPA for 6 (*n* = 3) and 24 (*n* = 12) hours (h). The tightness of the barrier remains intact after treatment with VPA for at least 24 h. (**C**–**E**) Expression of *Tfrc*, *Glut1*, and *CD98hc* mRNA. (**C**) The expression of *Tfrc* in BCECs increases after 6 h of treatment with VPA and returns to baseline after 24 h. In contrast, there are no changes in expression of *Glut1* (**D**) or *CD98hc* (**E**) following VPA treatment. Gene expression was analyzed using one-way ANOVA with Dunnett’s multiple comparisons test. Data (*n* = 4) are shown as mean ± SEM, *** *p* = 0.0001–0.001.

**Figure 3 cells-13-01181-f003:**
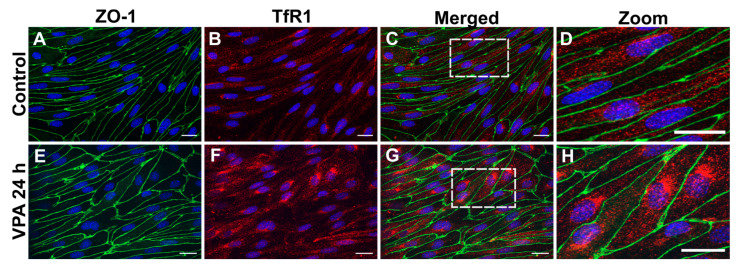
TfR1 protein in cultured brain endothelial cells (BCECs) after valproic acid (VPA) treatment. Zonula occludens (ZO-1; green) and TfR1 (red) detection using immunocytochemistry in BCECs co-cultured with mixed glial cells and treated with PBS (**A**–**D**) or 2 mM VPA (**E**–**H**). The tight junction protein ZO-1 remains highly expressed despite treatment with VPA (compare (**A**) with (**E**)). Morphologically, the increased expression of *Tfrc* (Figure 2) translates to increased TfR1 protein, compare (**B**) with (**F**). When shown in high magnification (**D**,**H**), it is evident that the TfR1 protein is highly expressed after VPA treatment, revealing a perinuclear distribution probably reflecting a higher abundance in the granular endoplasmic reticulum. The nuclei are stained with DAPI (blue). Scale bar: 20 μm. White box in (**C**,**G**) represents the zoomed in image (**D**,**H**).

**Figure 4 cells-13-01181-f004:**
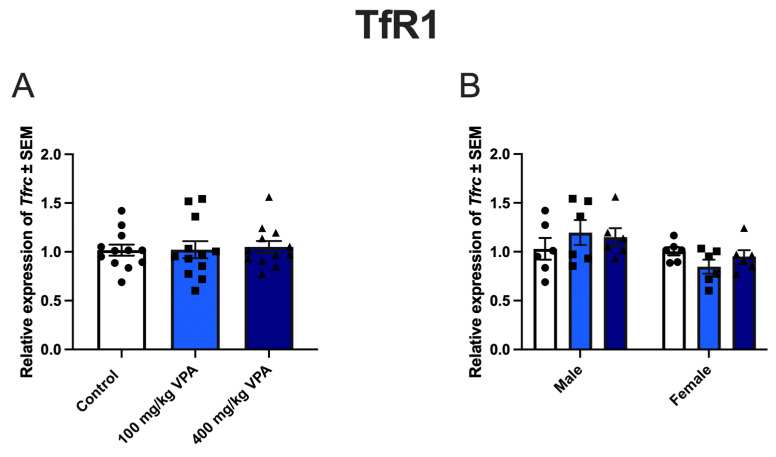
*Tfrc* gene expression (**A**,**B**) and TfR1 protein levels (**C**,**D**) after valproic acid (VPA) treatment in vivo; uptake of gold-labeled TfR1-targeted antibodies (**E**–**H**). (**A**,**B**) *Tfrc* gene expression in capillary-depleted brain extracts from mice treated with PBS, 100 mg/kg VPA, or 400 mg/kg VPA for 24 h. (**A**) Total cohort. (**B**) Separation by sex. (Males, *n* = 6; Females, *n* = 6 for each treatment group). Gene expression is analyzed using one-way ANOVA with Dunnett’s multiple comparisons and when investigating the difference in mRNA expression between male and female mice, a two-way ANOVA with the Holm–Sidak multiple comparisons test is used. Data (*n* = 6–12) are shown mean ± SEM. No differences are observed in *Tfrc* mRNA expression in the group of all animals (**A**) or after division by sex (**B**). (**C**) When analyzed for TfR1 protein in isolated capillaries, both 100 mg (*n* = 12) and 400 mg (*n* = 12) VPA treatment for 24 h significantly increased TfR1 protein, with upregulation in both male and female mice (**D**). One-way ANOVA with Dunnett’s multiple comparisons test was used for statistical analysis. ** *p* = 0.001–0.01. Data (*n* = 11–12) are depicted as mean ± SEM. (**E**–**H**) Uptake of gold-labeled TfR1-targeted antibodies. Uptake of TfR1-conjugated gold 1 nm nanoparticles in the brains, livers, spleens, and lungs of untreated mice and mice treated with 100 mg/kg VPA. The uptake of TfR1-antibody-conjugated gold nanoparticles, determined based on the content of gold in the isolated capillaries and brain fraction using ICP-MS, is similar in the brains (**E**) and hepatocytes of the liver (**F**) in control and VPA-injected mice. The uptake in the spleen (**G**) and lung (**H**) seems to increase slightly, although the differences are insignificant. The statistical difference between control and VPA-treated animals is analyzed with an unpaired *t*-test, each point represents data from a single mouse (*n* = 6), and the data are depicted as mean ± SEM.

**Figure 5 cells-13-01181-f005:**
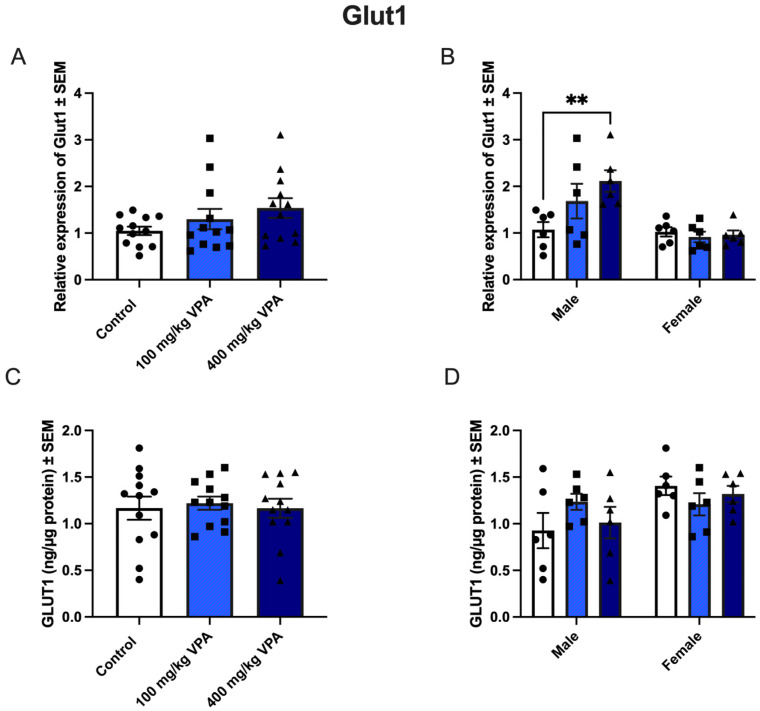
GLUT1 (**upper** panel) and CD98hc (**lower** panel) gene and protein expression in isolated brain capillaries after valproic acid (VPA) treatment. **Upper** panel. (**A**) Relative *Glut1* gene expression in brain capillaries from mice treated with PBS (*n* = 12), 100 mg/kg VPA (*n* = 12), or 400 mg/kg VPA (*n* = 12) for 24 h. (**B**) The *Glut1* expression profile is separated based on sex (males *n* = 6 and females *n* = 6 for each treatment group). No differences were observed in *Glut1* expression in the group of all animals (**A**), but significance was seen in the group of males when separated by sex (**B**) ** *p* = 0.001–0.01. Gene expression was analyzed using one-way ANOVA with Dunnett’s multiple comparisons test; when investigating the difference in gene expression between male and female mice, a two-way ANOVA with the Holm–Sidak multiple comparisons test was used. Data (*n* = 6–12) are shown as mean ± SEM, ** *p* = 0.001–0.01. (**C**,**D**) When GLUT1 protein was analyzed in isolated capillaries from controls (*n* = 12) and mice treated with 100 mg/kg VPA (*n* = 12) and 400 mg/kg VPA (*n* = 12), no significant difference in GLUT1 was seen, even when separated by sex (**D**). One-way ANOVA with Dunnett’s multiple comparisons test was used for statistical analysis. Data (*n* = 12) are depicted as mean ± SEM. (**E**,**F**) Relative *Cd98hc* gene expression in brain capillaries from mice treated with PBS (*n* = 12), 100 mg/kg VPA (*n* = 12), or 400 mg/kg VPA (*n* = 12) for 24 h. The *Cd98hc* expression profile was separated based on sex (males *n* = 6 and females *n* = 6 for each treatment group). Change in gene expression was analyzed using one-way ANOVA with Dunnett’s multiple comparisons test; when investigating the difference in gene expression between male and female mice, a two-way ANOVA with the Holm–Sidak multiple comparisons test was used. No differences were observed in *Cd98hc* expression in the group of all animals (**A**), nor when they were separated by sex (**B**). Data (*n* = 6–12) are shown as mean ± SEM. (**G**,**H**) CD98hc protein in the capillaries of mice injected with PBS (*n* = 12) or VPA in concentrations of 100 mg/kg (*n* = 12) or 400 mg/kg (*n* = 12) that was allowed to circulate for 24 h. The VPA treatment does not lead to significant changes in the CD98hc protein content. One-way ANOVA with Dunnett’s multiple comparisons test.

## Data Availability

The original contributions presented in the study are included in the article, further inquiries can be directed to the corresponding author.

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
