# Peer review of "Upregulation of Transferrin Receptor 1 (TfR1) but Not Glucose Transporter 1 (GLUT1) or CD98hc at the Blood–Brain Barrier in Response to Valproic Acid"

_cells, 2024, doi:10.3390/cells13141181_

Round 1
Reviewer 1 Report
Comments and Suggestions for Authors
-
Thank you for the opportunity to review the present article. This manuscript is well written. However, the author may take note of the major and minor remarks listed below to improve the manuscript:
Major comments:
- The introduction seems incomplete as it only focuses on TfR1 and not other studied receptors. I would request that authors include relevant literature in the introduction to emphasize the study hypothesis.
- I would like to see the TEER raw data. If possible, I request that authors provide it as supplementary data.
- Did the authors evaluate the intracellular ferratin levels in vitro and in vivo?
- Several other studies have shown contradictory results with valproic acid administration and glut1 expression in the brain and other organs. I would encourage authors to discuss these in the discussion section.
- Also, please include the study findings and future prospects in the discussion section.
Author Response
Reviewer 1
- The introduction seems incomplete as it only focuses on TfR1 and not other studied receptors. I would request that authors include relevant literature in the introduction to emphasize the study hypothesis.
Reply:
We appreciate the reviewer's suggestion to expand the introduction with more literature to emphasize the study hypothesis. Additional information on GLUT1 and CD98hc has been included as requested.
- I would like to see the TEER raw data. If possible, I request that authors provide it as supplementary data.
Reply: We believe we already provided both the raw data and a graph for the TEER results. On request, the reviewer is free to contact us directly for additional information on the BBB properties.
- Did the authors evaluate the intracellular ferratin levels in vitro and in vivo?
Reply: We did not evaluate intracellular ferritin levels in this study as the objectives were to investigate expression of potentially antibody-targetable molecules, but we agree it would have been interesting to dwell more on the iron metabolism in this model, which would take a separate study.
- Several other studies have shown contradictory results with valproic acid administration and glut1 expression in the brain and other organs. I would encourage authors to discuss these in the discussion section.
Reply: We agree on that there are conflicting observations reported previously. We have tried to expand the discussion on GLUT1 and VPA as suggested.
- Also, please include the study findings and future prospects in the discussion section
Reply: The study findings have been incorporated into the conclusion.
Reviewer 2 Report
Comments and Suggestions for Authors
The authors investigate the effect of valproic acid (VPA) treatment on the expression of 3 main targets for brain drug delivery, such as TfR1, GLUT1, and CD98hc using primary brain capillary endothelial cells (BCECs) co-cultured with mixed glia cells and C57BL/6JRj mice. Substantially, no changes are reported for GLUT1 and CD98hc, while increased expression of TfR1 is observed in both in vivo and in vitro models after VPA treatment. In addition, their data indicate that the uptake of gold-labeled anti-TfR1 antibodies is not altered in the brain, liver, spleen, and lung of VPA-treated mice.
The manuscript is interesting and well-written with an introduction that provides a clear and concise background of the topic. However, some revisions may improve it and help readers appreciate the study.
- How do the authors explain the increased expression of Tfrc in vitro after 6 hours of VPA treatment followed by a decrease after 24 hours of treatment?
- I suggest the authors confirm and quantify the increased expression of TfR1 protein observed in vitro using other methods for better comparison with the other results.
- Specify the choice of the VPA concentration used for uptake experiments.
- In literature there are other studies on the involvement of VPA in glucose metabolism in which significant changes in GLUT1 expression are reported both in vivo and in vitro. I suggest that the authors include these studies in the discussion by comparing their results with existing literature and explaining the differences and limitations.
- Finally the authors conclude that it is possible to upregulate TfR1 protein in BCECs using VPA. However they should clearly provide the conclusion. How do their results contribute to drug delivery field? What are the potential implications for future therapeutic strategies targeting TfR1-related pathways?
In conclusion, considering the quality of the paper I recommend the article to be accepted once the suggested modifications.
Author Response
Reviewer 2
The authors investigate the effect of valproic acid (VPA) treatment on the expression of 3 main targets for brain drug delivery, such as TfR1, GLUT1, and CD98hc using primary brain capillary endothelial cells (BCECs) co-cultured with mixed glia cells and C57BL/6JRj mice. Substantially, no changes are reported for GLUT1 and CD98hc, while increased expression of TfR1 is observed in both in vivo and in vitro models after VPA treatment. In addition, their data indicate that the uptake of gold-labeled anti-TfR1 antibodies is not altered in the brain, liver, spleen, and lung of VPA-treated mice. The manuscript is interesting and well-written with an introduction that provides a clear and concise background of the topic. However, some revisions may improve it and help readers appreciate the study.
- How do the authors explain the increased expression of Tfrcin vitro after 6 hours of VPA treatment followed by a decrease after 24 hours of treatment?
> Reply: A short explanation for the alteration in Tfrc mRNA status has been added.
I suggest the authors confirm and quantify the increased expression of TfR1 protein observed in vitro using other methods for better comparison with the other results
> Reply: We agree that an ELISA would have been optimal. However, due to ethical concerns regarding the large number of animals required for isolating BCECs in transwell BBB studies, we opted for an in vivo study instead of additional in vitro work.
- Specify the choice of the VPA concentration used for uptake experiments.
> Reply: The mice in the transport study were treated with 100 mg/kg VPA, and this has now been specified.
- In literature there are other studies on the involvement of VPA in glucose metabolism in which significant changes in GLUT1 expression are reported both in vivo and in vitro. I suggest that the authors include these studies in the discussion by comparing their results with existing literature and explaining the differences and limitations.
> Reply: We thank the reviewer for this valuable comment, and it has now been addressed.
- Finally the authors conclude that it is possible to upregulate TfR1 protein in BCECs using VPA. However they should clearly provide the conclusion. How do their results contribute to drug delivery field? What are the potential implications for future therapeutic strategies targeting TfR1-related pathways?
> Reply: We also thank the reviewer for these valuable suggestions. We have provided a clearer explanation as requested.
Round 2
Reviewer 1 Report
Comments and Suggestions for Authors
Thank you for adding the necessary changes.
Regards,
Namdev
Author Response
We thank for these comments.
Our replies:
1) We agree that the one time point approach looks quite simple. We did a similar experiment before to target the BBB using a high-affine anti-transferrin receptor nanoparticle, and we did not see further uptake even 24 hrs later in vivo [Johnsen et al. Sci Rep]; in that study we also examined uptake of the TfR1 targeted nanoparticles in vitro. Therefore we decided to study the effects of VPA at 60 min.
2) We have inserted a line as requested
3) We have rechecked the references and they are now adequate. The concrete reference 42 is correct by missed a publishers reference, as this was part of a PhD-thesis
Regards
Torben Moos